# Effects of service dogs on children with ASD's symptoms and parents' well-being: On the importance of considering those effects with a more systemic perspective

Nicolas Dollion[1]*, Margot Poirier[2], Florian Auffret[3], Nathe François[4], Pierrich Plusquellec[5,6], Marine Grandgeorge[2], Handi'Chiens[3¶], Fondation Mira[4¶]

1 Laboratoire C2S (Cognition Santé Société)–EA6291, Université Reims Champagne-Ardenne, Reims, France, 2 CNRS, EthoS (Éthologie animale et humaine)—UMR6552, Normandie Univ, Univ Rennes, Rennes, France, 3 Association Handi'chiens Inc., Paris, France, 4 Fondation Mira, Sainte-Madeleine, QC, Canada, 5 Centre d'études en sciences de la communication non verbale, Research Centre, Montréal Mental Health University Institute, CIUSSS Est, Montréal, Canada, 6 School of Psychoeducation, University of Montréal, Montréal, QC, Canada

¶ Membership of the author group with specific involvement in the present study can be found in the Acknowledgments
* nicolas.dollion@univ-reims.fr

**Data Availability Statement:** All relevant data are within the paper and its Supporting Information files.

## Abstract

The integration of a service dog can have numerous benefits for children with Autism Spectrum Disorder (ASD). However, although integration takes place within a family, little is known about the dynamics of these benefits on the family microsystem. Thus, the aim of our study was to propose a more systemic perspective, not only by investigating the benefits of SD integration, but also by exploring the relationships between improvements in children with ASD, parents' well-being, parenting strategies and the quality of the child-dog relationship. Twenty parent-child with ASD dyads were followed before, as well as 3 and 6 months after service dog integration. At each stage, parents completed an online survey which included: the Autism Behavior Inventory (ABI-S), the State-Trait Anxiety Inventory (STAI-Y), the Parenting Stress Index Short Version (PSI-SF), the Monash Dog Owner Relationship Scale (MDORS) and the Parenting Styles and Dimensions Questionnaire (PSDQ). First, repeated measure one-way ANOVAs revealed that both children's ASD symptoms and parents' anxiety decreased significantly after service dog integration. Additionally, Spearman correlations revealed that the more ASD symptoms decreased, the more parent's anxiety and parenting stress also decreased. Second, the quality of the child-dog relationship appeared to contribute to those benefits on both children's ASD symptoms and parents' well-being. Interestingly, parenting strategies seemed to adapt according to these benefits and to the quality of the child-dog relationship. Through a more systemic perspective, this study highlighted that the integration of a service dog involved reciprocal and dynamic effects for children with ASD and their parents, and shed new light on the processes that may underlie the effects of a service dog for children with ASD.

**Funding:** This study was funded by the Adrienne and Pierre Sommer Foundation, Handi'Chiens association, Mira Foundation and Brittany Region. This study was performed with a participatory action research approach. Thus, the Mira Foundation and the Handi'Chiens had a role in the development of the study protocol and design, in the recruitment of participant, the interpretation of final results and in the writing of the manuscript. They however had no role in the data collection and analysis, and decision to publish. Other funders had no role in study design, data collection and analysis, decision to publish, or preparation of the manuscript.

**Competing interests:** The authors have declared that no competing interests exist.

## 1. Introduction

Autism Spectrum Disorder (ASD) is a neurodevelopmental disorder with an early onset, causing lifelong difficulties. It is one of the most commonly diagnosed neurodevelopmental disorders. According to the most recent survey performed by the CDC, it is estimated that 1.85 up to 2.78 percent of children are currently diagnosed with ASD [1–3]. According to the most recent definition—DSM-V and CIM-11—[4, 5], ASD is characterized by a symptom dyad that includes: (A) deficits in social communication and social interaction, including impairments in both verbal (e.g., echolalia, delayed language development) and non-verbal communication (e.g., facial expressions, gestures), as well as difficulties engaging and maintaining social interaction and to respecting social interaction norms; and (B) restricted and repetitive behaviors, interests and activities, notably including both verbal and motor stereotypies, need for routine and immutability, and restricted fields of interests. These symptoms have lifelong consequences on the individual's daily functioning and quality of life [6–8].

In addition to the symptom dyad that characterizes ASD, the most recent version of the DSM [4] indicates that sensory atypicalities—usually referred to as sensory processing disorders—are also frequently present in individuals with ASD. These can translate into either sensory hypo-reactivity (i.e., obliviousness to stimulation), sensory hyper-reactivity (i.e., exaggerated response to stimuli) or sensory stimulation-seeking behaviors (i.e., sensory craving), which may all either affect specific senses or vary in intensity and polarity according to sensory modalities within a same individual [9–11]. Studies have shown that the sensory processing disorder of children with ASD can affect both their functional abilities (e.g., exacerbate the expression of stereotypies, increase the expression of atypical or self-injurious behaviors, decrease social participation) as well as their well-being (e.g., overarousal, anxiety, trigger avoidance or negative response) [12–16]. Additionally, individuals with ASD often have other difficulties and comorbidities that may add to their ASD. Approximately two thirds of children with ASD are diagnosed with at least one other comorbid disorder (e.g., anxiety, ADHD, epilepsy, intellectual disability) [17, 18]. These additional diagnoses will add to the range of difficulties and challenges that these children may face. Furthermore, not only do ASD deficits in social interaction and communication affect children's ability to interact and connect with their peers, but their atypical behaviors also affect peers' attitudes toward them and contribute to their ostracization [19, 20]. In particular, it has been shown that children with ASD are at higher risk of victimization and bullying than neurotypical (NT) children [21–23]. Thus, all these elements, either directly associated with ASD or adding to ASD symptomatology, may contribute to a reduced quality of life for these children, which may in turn have consequences for their families.

Indeed, parents of children with ASD have higher levels of stress (i.e., both general stress and parenting stress) and are at higher risks of psychological health issues (e.g., anxiety, depression) compared to parents of NT children [24–27]. In relation to these higher stress levels, disrupted cortisol activity has been reported in mothers of children with ASD (i.e., reduced morning cortisol activity and lower cortisol levels during the day) [28–30]. Different studies even show that parents' stress levels and their perceived quality of life (QoL) is related to both the severity of their child's ASD and its symptomatology (i.e., problematic behavior, IQ, distractibility) [31–33]. However, elements external to the child with ASD may also contribute to this higher level of stress and reduced QoL, such as feelings of social stigma (i.e., being blamed, feeling judged, feeling embarrassed) and social isolation [31, 34–38].

Parenting strategies correspond to behaviors, attitudes and practices sharing common traits that parents may use and express when they communicate and interact with their child. Based on their tendency to rely on specific parenting strategies rather than others, parents' parenting

style can be drawn [39]. Parenting styles can notably be defined according to two main dimensions (i.e., warmth and control), and four main parenting styles can be distinguished based on parents' positioning relatively to those dimensions: Authoritative [warmth+/control+], Authoritarian [warmth-/control+], Permissive [warmth+/control-] and Disengaged [warmth-/control-] [40–43]. The authoritative parenting strategies and style are generally considered to lead to more positive developmental outcomes for the child, compared to other parenting strategies and styles, such as for example, authoritarian parenting, which may lead to greater internalizing behaviors in children [44]. Various studies have shown that parental stress influences parental behaviors and parenting style [e.g., 45–48]. Thus, unsurprisingly, numerous studies show that parents of children with ASD show differences in their parenting strategies and styles compared to parents of NT children, [49, 50]. In terms of parenting behaviors, studies seem to show that parents of children with ASD use more material rewards, tend to adapt the environment more, provide less structure, are less sensitive (i.e., timing and quality of interventions), and rely less on rules and discipline, while relying more on positive parenting behaviors [e.g., 51–53]. However, there are conflicting findings across studies, which may be due in part to methodological differences. For example, in their meta-analysis, Ku and collaborators [49] report that while parents of children with ASD do not differ from parents of NT children in warmth and supportive behavior, they display more controlling (e.g., physical contact, imperative initiatives, unsynchronized comments) and negative behaviors (e.g., display anger, reject child's ideas) toward their child. In terms of parenting styles, the authoritarian style has been shown to be prevalent in parents of children with ASD compared to parents of NT children, while they rely less on authoritative strategies [27, 54, 55]. Interestingly, the higher level of stress experienced by parents of children with ASD appears to contribute to these differences in parenting strategies. In their study, Boonen and collaborators [51] demonstrated that while parents of children with ASD showed differences in their parenting behaviors compared to parents of NT children (i.e., lower scores on sensitivity and higher scores on material rewarding), these differences did not prove significant when statistically controlling for the effect of parenting stress.

Thus, considering both parental stress and parenting strategies simultaneously could be of critical interest, not only because of the impact of stress on parenting strategies, but even more importantly because of their potential deleterious effects on children with ASD—and because they are modifiable factors. Indeed, both parental stress and reliance on authoritarian and permissive parenting strategies have been shown to negatively impact the development and well-being of children with ASD [42, 56, 57]. Additionally, in their study, Osborne and collaborators [58] documented that parental stress can even affect the effectiveness of interventions provided to children with ASD.

Nowadays, numerous intervention strategies are proposed to improve the development and well-being of youths with ASD. One of these intervention strategies is the attribution of a service dog (SD). SDs for children with ASD are dogs that specifically selected and trained to assist these children in their daily life and throughout various activities, with the aim of improving their daily functioning and well-being [59, 60]. Studies on this topic show that the integration of a service dog can be the source of numerous benefits for children with ASD. In particular, it can lead to a reduction in ASD symptoms and severity, improvements in the child's emotional well-being (e.g., reduced stress, increased confidence, enhanced self-esteem), enhanced psychosocial development (e.g., improved social reciprocity, communication skills and prosocial behaviors) and a decrease in the expression of problematic behaviors (e.g., temper tantrums, sleep problems, stereotypies and running away) [61–70].

However, the benefits of integrating a SD within the family household are not limited to the child with ASD, as they may also extend to parents as well, even though they are not the

primary recipients of the SD. Specifically, following SD integration, parents of children with ASD experience improvement in their feelings of parental competence and security, as well as decreases in their stress and anxiety [62, 63, 67–69, 71, 72]. Using cortisol sampling, Fecteau and collaborators [29] demonstrated that while parents of children with ASD had low morning cortisol activity prior to SD integration, they showed a decrease in their parenting stress and an improvement in their cortisol activity after SD integration.

It has been speculated that the child's interaction and the quality of the relationship with the animal play a crucial role in the benefits that pets can provide for children with ASD [63, 73–76]. However, only a few studies have addressed this question, and these studies remain limited to pet dogs. In particular, they highlight that the quality of attachment between children with ASD and pet dogs is associated with the improvement of their social skills [74, 75, 77]. Addressing the attachment of children with ASD to their service dog may therefore be of interest, especially when considering that not all children with ASD become attached to their pets to the same extent—with some failing to develop a privileged relationship with the animal [74, 75, 78, 79]—and that not all children with ASD exhibit the same behavioral interaction profile when interacting with a SD [80, 81].

As for the parents, it has been hypothesized that the benefits of integrating a SD on their child with ASD contributes to improvements in their quality of life and well-being [29, 71, 72]. Based on parents' responses to an online survey which included several standardized scales, Carlisle and collaborators [74] observed that parents' perception of the benefits of having a pet animal for their child with ASD (i.e., dog and cat) was inversely related to their level of stress and positively related to the child's attachment to the pet, while the opposite was observed for the perceived burden of owning a pet. Interestingly, parents' stress was inversely related to their child's bond with the pet. Further to this last finding, various studies based on interviews and questionnaires show that parents may experience an additional burden due to the presence of a pet/SD in the family household or refer concerns about potential additional burden preventing them from integrating a pet/SD in the family household [76, 78, 82–86]. Non-optimal interaction between the child with ASD and the animal (i.e., either non-optimal behavior from the child [e.g., intrusive handling, excessive solicitation, probing] or from the dog [e.g., rough movements, excessive barking]), could lead to additional burden for parents; necessitating either additional caution and/or constant monitoring of child-dog interaction. While the association between child-dog relationship and parents' well-being have been previously examined in families of children with ASD with a pet dog [74], this association has never been examined in families integrating a SD. Investigating it may nevertheless be important, because although SDs are already fully trained, their integration entails additional responsibilities for the families of children with ASD (e.g., training routine, feeding, integration into almost every family activity, walks, inclusion in public outings) [63, 65, 87], which fall to the parents when their child has poor autonomy or attachment to his/her SD. Given that parents of children with ASD may already experience high levels of stress prior to SD integration, it is of even greater interest to consider this potential additional burden and stressor associated with poor child-SD relationship [29, 63, 85, 88].

It thus appears that integrating a SD into the family of a child with ASD may entangle numerous dynamic and reciprocal influences within the child-parent-SD trio that is formed, particularly between the child with ASD and his/her parent (e.g., the child's ASD symptoms have an influence on the parent's well-being and parenting strategies), and between the child with ASD and the SD (i.e., the quality of the child-dog relationship has an influence on benefits for the child), which may reciprocally influence each other (e.g., the benefits of SD on the child may lead to benefits on the parents which may lead to changes in how they interact with their child) [for a personal account, 89].

More generally, current knowledge regarding the impacts of ASD on both the child's and parent's quality of life, as well as on the effects of SDs on children with ASD and their parents, seems to point to the importance of considering the effects of ASD and service dogs with a more systemic perspective, at least within the microsystem. Simultaneously considering the various aspects and subsets of effects involved in the integration of a SD in the family of a child with ASD (i.e., outcomes on the child's ASD, parents' stress and anxiety, parenting strategies toward the child, child-dog relationship) and their interrelationships could help improve our understanding of the benefits of SD on youths with ASD and their families, and the complex dynamics behind these effects. Furthermore, considering these effects with a more systemic perspective is all the more relevant, as the SD integrates a family microsystem and becomes a member of that system, and can even be considered as a member of the family [90].

Therefore, the main objective of the present study was to propose a more systemic perspective on the benefits of SD integration for families of children with ASD. Based on current knowledge, the specific aims of the study were:

- **Aim#1:** To support previous studies, by confirming that parental well-being (parenting stress and anxiety) is related to their child's ASD symptoms [Aim#1A] and that significant improvements are observed both in the child's symptoms and the parents' well-being following SD integration [Aim#1B].

- **Aim#2:** To investigate whether changes in children's ASD symptoms following the integration of a SD are associated with changes in parental well-being.

- **Aim#3:** To examine whether changes in children's ASD symptoms and in parental well-being vary in relation to the quality of the child-dog relationship.

- **Aim#4:** To explore how parents' parenting strategies toward their child adapted within this context by (Aim#4A) determining whether their parenting strategies were related to their child's quality of relationship with the service dog, and by (Aim#4B) examining whether their parenting strategies adapted according to the benefits of SD integration for both their child with ASD and their own well-being.

## 2. Material and methods

### 2.1. Ethics

The present study was approved by the French Ethical Committee (French CPP; RCB number: 2020-A02012-37) and the Ethics Committee for Educational and Psychological Research (CEREP) of the University of Montréal (Certificate number: CEREP-20-113-D). This study is based on a longitudinal design consisting of surveys completed by parents of children with ASD at three different stages (i.e., baseline survey before SD integration, and 2 follow-ups surveys 3 and 6 months after). Its methodology was thus completely non-invasive and was conducted in agreement with the principles of the Declaration of Helsinki, revised in 2000. All parents provided their written consent before completing each survey (i.e., one written consent prior to each survey completion). All participating families were recruited between December 2020 and October 2022. All data were denominated.

### 2.2. Participants

Participants in this study were dyads of parents of children with ASD and their child. The initial sample consisted of 35 dyads. However, from this initial sample, 14 dyads dropped out of the study and thus did not complete data collection at all three stages of assessment. Some of

the parents did not provide a reason for dropping out (i.e., no response to reminder emails), while others stated that they were unable to complete the online survey or that they had to discontinue their participation for at least one of the three stages of assessment, due to circumstances (e.g., job change, busy schedule, illness of a family member). In addition, one family had to return the service dog prior to the 6-month follow-up, due to inappropriate behavior of the child with the service dog (i.e., intrusive exploratory behaviors, excessive solicitations, etc.). As a result, the parent was unable to complete the final survey. Any data collected from those families who did not complete the baseline survey and both follow-up surveys were not included in the final dataset used for analysis. Hence, the analyzed sample comprised a total of 20 dyads.

All dyads were recruited from the waiting lists (i.e., families selected to receive a service dog) of the Mira Foundation and the Handi'Chiens Association. The Mira Foundation (www. mira.ca), located in Quebec (Canada), and the Handi'Chiens Association (www.handichiens. org), located in France, are two non-profit organizations that train and donate service dogs to individuals with various forms of handicaps and disabilities, including children with ASD and their families. For recruitment, the following inclusion criteria were applied: (a) the child had to be diagnosed with ASD (i.e., diagnosis provided by a clinician), (b) the child had to be between 5 and 12 years of age (i.e., between 5y0mo and 12y11mo) at the time of SD attribution, (c) the child had to live at home with his/her parent(s); and (d) the family had to be about to receive a SD in the following weeks from either the Mira Foundation or the Handi'Chiens Association.

The final sample included twenty parents (19 mothers and 1 father) and their child with ASD (7 girls and 13 boys) who participated in this study at all three stages of assessment. All children had a diagnosis of ASD delivered by a professional (mean score on the Social Communication Questionnaire [SCQ] of 21.1±5.5, see information about the scale in section 2.3.a.), and among these 20 children with ASD, 13 had one or more additional comorbidities (e.g., ADHD, generalized anxiety disorder, epilepsy, emotional regulation disorder, see Table 1). The mean age of the parents at the time of SD attribution was 41.0±4.1 years old, while the mean age of the children with ASD was 111.0±25.3 months old. Nine dyads were from the province of Quebec (Canada) and received a SD trained and delivered by the Mira Foundation, while eleven dyads were from France and received a SD trained and provided by the Handi'Chiens Association. The SDs were of different breeds (i.e., 9 Labernese, 4 Labrador, 6 Golden Retrievers, 1 German Shepherd) and their mean age at attribution was 24.5±3.6 months old.

## 2.3. Methods

**2.3.a. Procedure.**   For this study, participants were recruited from the waitlists of both the Mira Foundation and the Handi'Chiens Association. The staff of both non-profit organization were instructed to check, among the families in their waitlists, for families of children with ASD who met the inclusion criteria and were about to be invited to participate in their next SD attribution session. Fulfillment of the inclusion criteria was verified on the basis of the documentation submitted by the families to the organizations when they registered as candidates for a SD—which included the child's full medical record—. All families who met the inclusion criteria and who confirmed their willingness to attend to the next attribution session scheduled by the organization were asked by the staff if they were interested in participating in this study. Only upon confirmation of their interest was the family put in contact with the research team by means of a contact email.

Interested parents were contacted by the research team in order to arrange an initial meeting which could be conducted either by video conference or by phone. The purpose of this

**Table 1. General information about the dyads (i.e., parents and their child with ASD) who participated in the study.**

| Dyad number | Child's sex | Child's age at SD attribution (in months) | Child's comorbidities | Parent's sex | Parent's age at SD attribution (in years) | SD provider | Family type | Delay between T0-SD attribution (in months) | Delay between SD attribution-T1 (in months) | Delay between SD attribution-T2 (in months) |
|---|---|---|---|---|---|---|---|---|---|---|
| 1 | M | 126 | ADHD | F | 43 | M | Nuc | 1,0 | 3,4 | 5,6 |
| 2 | M | 97 | - | F | 47 | M | Nuc | 0,8 | 3,1 | 5,9 |
| 3 | M | 138 | ADHD—AD | F | 37 | M | Nuc | 1,4 | 3,3 | 6,3 |
| 4 | F | 134 | AD | F | 42 | M | Nuc | 1,4 | 2,5 | 6,0 |
| 5 | F | 150 | - | F | 41 | M | Nuc | 1,6 | 2,7 | 5,6 |
| 6 | F | 120 | AD | F | 43 | M | Mon | 1,6 | 2,8 | 6,1 |
| 7 | M | 134 | ADHD -LDD—IDD- MDD- X | F | 46 | M | Mon | 1,5 | 3,5 | 6,7 |
| 8 | M | 88 | ADD -LDD | F | 35 | M | Nuc | 1,5 | 2,6 | 6,3 |
| 9 | F | 150 | - | F | 38 | M | Nuc | 1,5 | 4,3 | 6,4 |
| 10 | F | 99 | - | F | 40 | HC | Nuc | 0,9 | 2,6 | 6,1 |
| 11 | M | 93 | LDD | F | 35 | HC | Nuc | 0,6 | 2,8 | 5,7 |
| 12 | M | 104 | MDD | F | 42 | HC | Nuc | 0,5 | 2,9 | 5,6 |
| 13 | M | 74 | - | F | 42 | HC | Nuc | 1,0 | 2,6 | 7,1 |
| 14 | F | 102 | ADHD—DYSL-LDD | F | 32 | HC | Mon | 1,9 | 3,2 | 5,9 |
| 15 | M | 88 | ADHD -LDD—ND- MDD—EP | F | 35 | HC | Rec | 0,6 | 2,8 | 6,0 |
| 16 | F | 127 | ADHD -OD—AD—DYSL—DYSO-DYSP | F | 45 | HC | Mon | 0,8 | 3,1 | 6,1 |
| 17 | M | 83 | - | M | 43 | HC | Nuc | 0,3 | 2,7 | 5,9 |
| 18 | M | 144 | ADHD | F | 45 | HC | Nuc | 1,2 | 2,7 | 5,8 |
| 19 | M | 79 | DYSP | F | 40 | HC | Nuc | 1,0 | 2,9 | 5,9 |
| 20 | M | 87 | - | F | 40 | HC | Nuc | 0,7 | 2,7 | 5,7 |

AD = Anxiety Disorder; ADD = Attention Deficit Disorder; ADHD = Attention Deficit Hyperactivity Disorder; DYSL = Dyslexia; DYSO = Dysorthography; DYSP = Dyspraxia; EP = Epilepsy; IDD = Intellectual Development Disorder; LDD = Language Development Disorder; MDD = Motor Development Disorder; ND = Neurovisual Disorder; OD = Oppositional Disorder; X = X Fragile Syndrome; M = Mira Foundation; HC = Handi'Chiens Association; Nuc = Nuclear family; Sing = Single-parent family; Rec = Recomposed family

initial meeting was first to provide parents with all necessary information about the study and answer any questions they might have prior to signing the informed consent, and then to collect family demographic information (i.e., child's age and sex, parent's age and sex, date of attribution session, contact information, confirmation of child's diagnosis and comorbidities, household composition).

A longitudinal study design was used in which parents completed an online survey at three key stages: before SD attribution (T0), 3 months after SD attribution (T1) and 6 months after SD attribution (T2). The online survey was conducted using LimeSurvey@. For the baseline assessment before SD attribution (T0), parents received the web link to the survey and could complete it between 1.5 months and 2 weeks prior to the SD attribution session (i.e., mean delay of 1.1±0.4mo). For the follow-ups at 3 and 6 months after SD attribution (T1 and T2), parents received the web link 2 weeks prior to the 3- and 6-month anniversaries of attribution, and were given up to 2 weeks after the anniversary date to complete the survey (i.e., mean delay of 3.0±0.4mo and 6.0±0.4mo respectively). However, for three parents who encountered special circumstances at one of the stages of assessment(i.e., 1 parent for T1, and 2 parents for

T2), the completion deadline was extended in order to adapt to their current situation (i.e., from 6 to 24 additional days).

At each stage of assessment, the online survey always included the same four standardized scales and questionnaires: the Autism Behavior Inventory Short Form (ABI-S), the Parenting Stress Index Short Form (PSI-SF), the State and Trait Anxiety Inventory Form Y (STAI-Y) and the Parenting Styles and Dimensions Questionnaire (PSDQ) (additional information relative to the standardized scales and questionnaires used [Cronbach alphas, goals, number of items, types of response, extracted scores, time to complete and stage it was used in] are available in supporting information S1 Table). The ABI-S [91] is a scale that allows for the measurement of changes in core and associated symptoms of ASD, and is used to assess the effectiveness of interventions. It includes 24 items, with Likert scale responses, covering five different domains: social communication, restrictive and repetitive behaviors, mental health, self-regulation and challenging behaviors. The PSI-SF [92] is a standardized scale for measuring parenting stress, in both clinical and research contexts. It includes 36 items, consisting of statements to which parents must indicate their degree of agreement on a five-point Likert scale. It includes three subscales: parental distress, parent-child dysfunctional interaction and difficult child. Its recommended clinical threshold cut-off for the total score is $\geq 90$. The STAI-Y [93] is a 40-item Likert-type rating scale designed to measure state and trait anxiety in participants. Using a four-point Likert scale, participants must indicate their degree of agreement with 20 statements regarding how they currently feel [state anxiety subscale–STAI-Y-1] and 20 statements regarding how they usually feel (i.e., over the past few weeks) [trait anxiety subscale–STAI-Y-2]. For each subscale, the recommended clinical threshold cutoff is $\geq 40$. The PSDQ [94, 95] is a 62-item questionnaire designed to assess parent's childrearing behavior and strategies according to the three major parenting styles defined by Baumrind [40]: authoritative (i.e., warm/involved, reasons/induces, encourages democratic participation, good-natured/easygoing), authoritarian (i.e., verbally hostile, uses corporal punishment, does not reason, uses punitive strategies, directive) and permissive (i.e., lacks follow-through, ignores misbehavior, lacks self-confidence). Each item describes a parenting practice or behavior, and parents are asked to provide a score from 1 to 5 to indicate how often they display the described behavior.

In addition, the baseline survey at T0 included the Social Communication Questionnaire [SCQ; 96]. This screening tool is used to assess communication skills and social functioning difficulties that may be indicative of ASD. Here, the Lifetime version of the tool was used, which looks at the child's entire developmental history. It includes 40 items consisting of closed-ended yes/no questions asking parents whether or not their child exhibits specific behaviors. This tool was used to characterize the ASD profile of the children in our sample (Table 1), and in particular to obtain a score indicating an approximate level of ASD severity (recommended clinical threshold cutoff score $\geq 15$ [96]).

The Monash Dog-Owner Relationship Scale [MDORS; 97] was included in the survey only for the T1 and T2 follow-ups (i.e., after SD attribution). The MDORS is a 28-item Likert-type rating scale that was initially used to measure the owner's perceived relationship with their dog (either SD or pet dog). It measures three different dimensions of the human-dog relationship: owner-dog interaction (*e.g.*, "How often do you hug your dog?"), perceived emotional closeness (*e.g.*, "I wish my dog and I never had to be apart") and perceived costs (*e.g.*, "My dog costs too much money"). Items consist either of questions or statements about one's actions or feelings toward his/her dog, and participants must indicate either how often they display the described behavior or how strongly they agree with the statement, using 5-point Likert scales. In its original version, the MDORS includes items referring to "I" or "me" (i.e., the dog's owner). However, for the purposes of the present study, the items were reformulated to refer

to "my child", in order to assess the quality of the child-dog relationship from a third-person perspective (i.e., his/her parent). This adjustment was made because ASD may particularly affect children's communication skills—with a proportion of individuals with ASD being non-verbal—which would thus interfere with the ability of children with ASD to directly complete the MDORS in its original version, either in a verbal or a written form.

**2.3.b. Data analyses.** Correlational analyses (i.e., Pearson correlation tests) and analyses of variance (one-way repeated measures ANOVAs) were used to analyze the data. The significance threshold was set at 0.05. All statistical analyses were performed using IBM SPSS Statistics version 28.0 (IBM, Inc., Armonk, NY, USA).

For Aim#1A, Pearson correlation tests were used at each stage of assessment (i.e., T0, T1 and T2) to determine whether there was an association between children's ASD symptoms (i.e., ABI-S total score) and their parent's stress and anxiety levels (i.e., PSI-SF total score, STAI-Y-1 and Y-2). According to Austin and Steyerberg [98], linear regression models require only two subjects per variable for an adequate estimation of regression coefficients, standard errors and confidence intervals. Thus, in order to further explore the relationship between children's ASD symptoms and their parent's stress and anxiety, linear hierarchical regressions were performed only on variables exhibiting significant correlations. Distinct hierarchical linear regressions were performed in two steps with ABI-S, STAI-Y-1 and STAI-Y-2 scores as dependent variables. In step one, potentially confounding variables related to the child with ASD (i.e., age and sex) were introduced. In step two, ABI-S scores were entered using a step-wise forward method. For Aim#1B, one-way repeated measures ANOVAs, followed by pairwise comparisons, were used to examine whether significant differences/improvement in the child's symptoms (i.e., ABI-S score) and in the parents' stress and anxiety (i.e., PSI-SF total score, STAI-Y-1 and Y-2 scores) could indeed be observed between the three stages of assessment, particularly before *versus* after SD attribution.

Since the next aims (#2, #3 and #4) involved exploring the associations of different variables with the changes in children's symptoms and parents' well-being (i.e., anxiety and parenting stress), change scores were calculated for the corresponding variables (i.e., PSI-SF total score, STAI-Y-1 and Y-2 scores). These change scores were calculated by simply subtracting the participants' baseline scores on each of the corresponding scales before SD attribution from the participants' follow-up scores after SD attribution. Change scores at 3 months were calculated by subtracting scores at T0 from scores at T1, and change scores at 6 months were calculated by subtracting scores at T0 from scores at T2.

For Aim#2, Pearson correlation tests were used to examine whether at T1 and T2, changes in children's symptoms (i.e., change in scores T1-T0 and T2-T0 on the ABI-S) were associated with changes in parents' anxiety and parenting stress (i.e., change in scores T1-T0 and T2-T0 on the STAI-Y-1 and Y-2, and on the PSI-SF). Similarly to Aim#1, distinct hierarchical linear regressions were applied afterwards for each variable association that manifested significant correlation in order to explore whether change scores in children's ASD symptoms predicted changes in parents' anxiety and parenting stress. The same two-step design was used (i.e., step 1: introduction of child's age and sex; step 2: introduction of ABI-S change scores).

Then, for Aim#3, to examine whether changes in children's symptoms and parents' well-being were associated with the child-dog relationship, Pearson correlation tests were used. To examine these associations, change scores (i.e., T1-T0 and T2-T0) for the ABI-S, the STAI-Y and the PSI-SF, as well as scores at T1 and T2 on each of the three dimensions of the MDORS (i.e., owner-dog interaction, perceived emotional closeness and perceived costs), were used. Analyses were performed by examining the correlations between change scores in children's symptoms and parents' well-being, with the MDORS scores at the corresponding stage of assessment (i.e., MDORS at T1 with T1-T0 change scores, MDORS at T2 with T2-T0 change

scores). Following the correlations test, the same hierarchical linear regression tests were applied with a strategy similar to Aims #1 and #2 (i.e., introduction of child's sex and age at step 1 and introduction of scores on dimensions of the MDORS at step 2) in order to test whether scores on dimension of the MDORS predicted change scores for children's symptoms and parents' anxiety and parenting stress. Here as well, distinct hierarchical regressions were performed only on variables whose correlation was significant.

Pearson correlation tests were used for the last two aims which focused on the effects of SD integration on parenting strategies. Correlations between scores on the three parenting subscales of the PSDQ (i.e., authoritarian, authoritative and permissive) and scores on the three dimensions of the MDORS were used to examine whether parenting strategies were associated with the quality of the relationship between child and service dog at each follow-up after SD integration (Aim#4A). Finally, to examine whether parenting strategies adapted according to the benefits of SD integration on the child with ASD and on parent's well-being (Aim#4B), we examined whether change scores for the three subscales of the PSDQ correlated with change scores for the ABI-S, PSI-SD, STAI-Y-1 and Y-2. These correlation analyses were performed using the two change scores on the subscales of the PSDQ (i.e., T1-T0 and T2-T0) with the corresponding change scores on the other considered scales (i.e., correlations between T1-T0 change scores for authoritative, authoritarian, permissive subscales with T1-T0 change scores for the ABI-S, PSI-SF, STAI-Y-1 and Y-2; and similarly, for T2-T0). As with previous aims, hierarchical linear regressions were applied to further explore correlations that proved significant, with the introduction of the child's sex and age at step 1. For Aim#4A, scores on dimensions of the MDORS were introduced at step 2 in order to test for their prediction of scores on PSDQ subscales. For Aim#4B, scores on dimensions of the ABI-S, PSI-SF, STAI-Y-1 and -2 were introduced at step 2 in order to test whether they predicted change scores on parenting strategies. Distinct hierarchical linear regression tests were performed for each scale and each stage of assessment.

## 3. Results

Participants' raw scores on the standardized scales and questionnaires at each stage of assessment, as well as mean scores and standard deviations, are available in supporting information S2 Table.

For the sake of clarity, for all hierarchical linear regression analyses performed (i.e., Aim#1 to Aim#4B), only results corresponding to the second step of the model will be reported, since the first step was performed to control for the effect of potential confounding variables (i.e., child's age and sex).

### 3.1. Association between child's ASD symptoms and parent's well-being (Aim#1A)

Correlation analyses of total ABI-S scores and PSI-SF and STAI-Y1/2 scores (Table 2) showed that: (a) ABI-S scores were positively correlated with PSI-SF scores at all three stages, (b)

**Table 2. Results of Pearson correlation tests performed on scores from scales measuring child's ASD symptoms [ABI-S] and parent's well-being [PSI-SF and STAY-Y] at each stage.**

|  | ABI-S with PSI-SF | ABI-S with STAI-Y State | ABI-S with STAI-Y Trait |
|---|---|---|---|
| Before SD (T0) | **.686 (p < .001)** | .040 (p>.05) | .122 (p>.05) |
| 3M after SD (T1) | **.787 (p < .001)** | **.530 (p = .016)** | .387 (p>.05) |
| 6M after SD (T2) | **.760 (p < .001)** | **.512 (p = .021)** | .405 (p>.05) |

ABI-S scores were positively correlated with scores on the STAI-Y State only at stages following SD attribution (i.e., T1 and T2), (c) ABI-S scores did not correlate with scores on the STAI-Y Trait at any of the three stages. In other words, higher levels of ASD symptoms were associated with higher levels of parenting stress at all three stages, and with higher levels of parent state anxiety after SD attribution.

After controlling for the effects of child with ASD's age at SD attribution and sex (step 1), introducing ABI-S scores at T0 (step 2) in the model explained an additional 45% of the variance in PSI-SF scores at T0 (F(1, 16) = 14.04), with ABI-S score being a significant predictor ($\beta$ = 0.688, p = .002). Similar hierarchical regression analyses performed at T1 and T2 revealed that, at both stages, ABI-S scores were significant predictors of PSI-SF scores (respectively, $R^2$ = 0.591, F(1, 16) = 27.01, $\beta$ = 0.778, p < .001; and $R^2$ = 0.594, F(1, 16) = 23.41, $\beta$ = 0.778, p < .001). Hierarchical linear regressions also revealed that ABI-S scores at T1 and T2 were significant predictors of STAI-Y-1 scores at corresponding stages (respectively, $R^2$ = 0.214, F(1, 16) = 5.14, $\beta$ = 0.477, p = .038; and $R^2$ = 0.233, F(1, 16) = 5.58, $\beta$ = 0.487, p = .031).

## 3.2. Changes in child's ASD symptoms and parent's well-being after SD attribution (Aim#1B)

One-way repeated measures ANOVA performed on the ABI-S scores at the three stages of assessment revealed a significant effect of time (Wilks' Lambda = 0.444, F(2, 18) = 11.26, p < .001). Pairwise comparisons revealed a significant difference between ABI-S scores at T2 (M = 61.4±23.43) compared to scores at T0 (M = 74.7±23.73, p = .01) and T1 (M = 66.8±23.21, p = .007) (Fig 1A).

Similar analyses for the PSI-SF scores revealed a significant effect of time as well (Wilks' Lambda = 0.695, F(2, 18) = 3.94, p = .038). Pairwise comparisons only revealed a significant difference between scores at T0 (M = 113.3±16.86) and scores at T2 (M = 104.4±19.40, p = .036), while other comparisons were not significant (all p>.05) (Fig 1A).

Likewise, one-way repeated measures ANOVA on the STAI-Y-2 scores [Trait anxiety] also revealed a significant effect of time (Wilks' Lambda = 0.644, F(2, 18) = 4.96, p = .019). Only scores at T0 and T2 differed significantly (respectively, M = 49.15±11.68 and M = 42.95±12.01, p = .014, while other comparisons did not yield significant results (all p>.05) (Fig 1B).

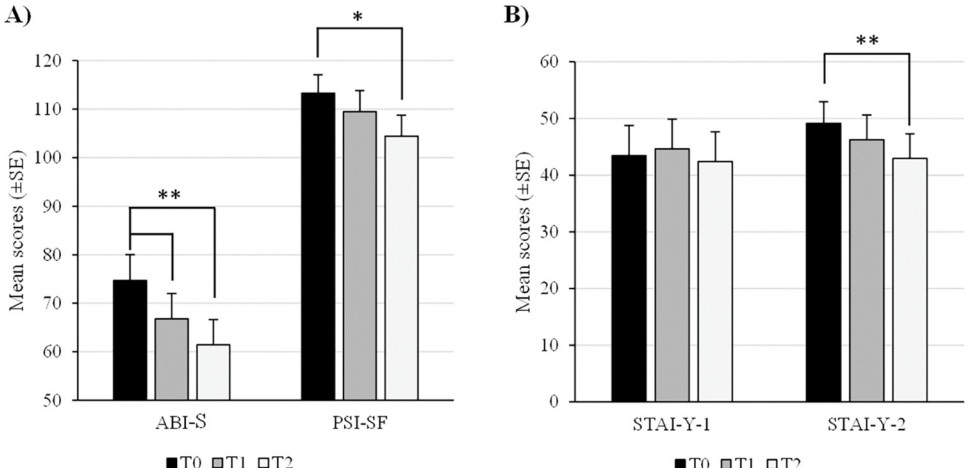

**Fig 1.** Participants' mean scores (±SE) on the ABI-S and the PSI-SF (A), and on the STAI-Y-1 and Y-2 (B) at each of the three stages.

Finally, one-way repeated measures ANOVA on the STAI-Y-1 scores [State anxiety] revealed no significant effect of time (Wilks' Lambda = 0.952, F(2, 18) = 0.45, p>.05) (Fig 1B).

These results show significant decreases in children's ASD symptoms 3 and 6 months after SD integration, as well as significant decreases in parents' parenting stress and trait anxiety 6 months after SD integration.

### 3.3. Association between changes in child's ASD symptoms and parent's well-being (Aim#2)

Tests performed on change scores between before SD attribution and 3 months after revealed that change scores on the ABI-S correlated positively with change scores on the PSI-SF (r = .590, p = .006) and on the STAI-Y-2 (r = .549, p = .012), whereas the correlation with change scores on the STAI-Y-1 was not significant (r = .309, p>.05). Similarly, for change scores between before SD attribution and 6 months after, change scores on the ABI-S correlated positively with change scores on the PSI-SF (r = .546, p = .013) and with change scores on both STAI-Y-1 and Y-2 (respectively, r = .481, p = .032; r = .592, p = .006). Thus, the more the child's ASD symptoms improved following SD attribution, the more the parent's parenting stress and anxiety also decreased.

Hierarchical linear regression performed with ABI-S change scores for the prediction of change scores on the PSI-SF and STAI-Y-2 between before SD attribution and 3 months after revealed that change scores on the ABI-S was a significant predictor of change scores on the PSI-SF ($R^2$ = 0.330, F(1, 16) = 8.33, $\beta$ = 0.667, p = .01), while only a tendency to predict change scores on the STAI-Y-2 was observed ($R^2$ = 0.168, F(1, 16) = 8.14, $\beta$ = 0.476, p = .065). All three hierarchical regression models performed on change scores between before SD attribution and 6 months after revealed that change scores on the ABI-S was a significant predictor of change scores on the PSI-SF, SATI-Y-1 and STAI-Y-2 ($R^2$ from 0.278 to 0.320, F(1, 16) from 6.69 to 7.99, $\beta$ from 0.583 to 0.624, all p < .05).

### 3.4. Association of child-SD relationship with changes in child's ASD symptoms and parent's well-being (Aim#3)

Correlations between scores on the three dimensions of the MDORS at T1 and change scores on the ABI-S, PSI-SF and STAI-Y1/2 between before SD attribution and 3 months after, revealed only a significant negative correlation between emotional closeness scores at T1 and changes in the ABI-S scores (r = -.444, p = .05) (Table 3). This indicates that the more children had an emotionally close relationship with their SD, the greater a decrease in their ASD

**Table 3. Results of Pearson correlation tests on change scores on scales measuring child's ASD symptoms [ABI-S] and parent's well-being [PSI-SF and STAY-Y], with scores on dimensions of the scale measuring child-SD relationship [MDROS] at each stage.**

| | | Before SD attribution-3M After (T1-T0) | | | |
|---|---|---|---|---|---|
| | | ABI-S | PSI-SF | STAI-Y State | STAI-Y Trait |
| 3M After SD Attrib. (T1) | MDORS -Closeness | **-.444 (p = .05)** | -.327 (p>.05) | -.230 (p>.05) | -.171 (p>.05) |
| | MDORS-Costs | -.358 (p>.05) | -.131 (p>.05) | .009 (p>.05) | -.104 (p>.05) |
| | MDORS-Interaction | -.326 (p>.05) | -.096 (p>.05) | .153 (p>.05) | -.146 (p>.05) |
| | | Before SD attribution-6M After (T2-T0) | | | |
| | | ABI-S | PSI-SF | STAI-Y State | STAI-Y Trait |
| 6M After SD Attrib. (T2) | MDORS- Closeness | **-.626 (p = .003)** | **-.496 (p = .026)** | -.235 (p>.05) | -.243 (p>.05) |
| | MDORS- Costs | -.407 (p>.05) | **-.524 (p = .018)** | -.358 (p>.05) | -.142 (p>.05) |
| | MDORS- Interaction | **-.493 (p = .027)** | -.350 (p>.05) | -.075 (p>.05) | -.176 (p>.05) |

symptoms was observed. All other correlations were not significant. The second step of the hierarchical linear regression revealed that scores on closeness dimension of the MDORS at 3 months was a significant predictor of change scores on ABIS-S between before SD attribution and 3 months after ($R^2 = 0.213$, F(1, 16) = 6.49, $\beta = -0.473$, p = .02).

Similar analyses conducted with scores on the three dimensions of the MDORS at T2 and change scores between before SD attribution and 6 months after revealed significant negative correlations between emotional closeness scores at T2 and changes in both the ABI-S and the PSI-SF scores (respectively, r = -.662, r = -.496, both p < .05) (Table 3). Additionally, perceived costs scores at T2 correlated negatively with changes in PSI-SF scores (r = -.524, p < .05), while child-SD interaction scores at T2 correlated negatively with changes in ABI-S scores (r = -.493, p < .05). Other correlations were not significant.

Thus, it appears that, particularly 6 months after SD attribution, the more the child was emotionally close and interacted with the SD, the greater was the decrease in his/her ASD symptoms, while the more the child was emotionally close and had a costless relationship with the SD, the greater was the decrease in his/her parent's parenting stress.

Hierarchical linear regression testing for the prediction of change scores on the ABIS-S between before SD attribution and 6 months after with scores on emotional closeness and child-SD interaction dimensions of the MDORS at 6 months led to emotional closeness being the only variable retained in the second step of the model as a significant predictor ($R^2 = 0.306$, F(1, 16) = 9.54, $\beta = -0.566$, p = .007), while the child-SD interaction score was not ($\beta = -0.134$, p>.05). Similarly only emotional closeness revealed to be a significant predictor of change scores T2-T0 in parenting stress ($R^2 = 0.253$, F(1, 16) = 5.75, $\beta = -0.515$, p = .03), while the cost of the relationship was not ($\beta = -0.299$, p>.05).

## 3.5. Association between parent's parenting strategies and child-SD relationship (Aim#4A)

Correlation analyses conducted on scores at T1 on dimensions of the MDORS and the PSDQ revealed that scores on the authoritative subscale of the PSDQ were positively associated with scores on the emotional closeness and child-SD interaction dimension of the MDORS (respectively, r = .447, r = .451, both p < .05, Table 4). Additionally, scores on the authoritarian subscale of the PSDQ were negatively correlated with scores on both the perceived costs and the child-SD interaction dimensions of the MDORS (respectively, r = -.614, r = -.460, both p < .05). Similarly, analyses at T2 revealed a significant negative correlation between scores on the authoritarian dimension of the PSDQ and scores on all three dimensions of the MDORS (r from -.464 to -.747, all p < .05). Thus, a higher quality child-SD relationship was associated with an increase in parents' use of authoritative strategies at T1 and a concurrent decrease in their use of authoritarian strategies at both T1 and T2.

**Table 4. Results of Pearson correlation tests performed on scores on dimension on scales measuring parenting strategies [PDSQ] and child-service dog relationship [MDROS] at follow-ups after SD attribution.**

| | 3M After SD attribution (T1) | | | | 6M After SD attribution (T2) | | |
|---|---|---|---|---|---|---|---|
| | PSDQ -Authoritative | PSDQ -Authoritarian | PSDQ -Permissive | | PSDQ -Authoritative | PSDQ -Authoritarian | PSDQ -Permissive |
| MDORS -Closeness (T1) | **.447 (p = .048)** | -.286 (p>.05) | -.188 (p>.05) | MDORS -Closeness (T2) | .099 (p>.05) | **-.464 (p = .039)** | -.254 (p>.05) |
| MDORS -Costs (T1) | .355 (p>.05) | **-.614 (p = .004)** | -.065 (p>.05) | MDORS -Costs (T2) | .110 (p>.05) | **-.747 (p < .001)** | -.196 (p>.05) |
| MDORS-Interaction (T1) | **.451 (p = .046)** | **-.460 (p = .046)** | -.280 (p>.05) | MDORS -Interaction (T2) | .080 (p>.05) | **-.555 (p = .011)** | -.249 (p>.05) |

Hierarchical linear regression revealed that the child-SD interaction dimension at T1 was a significant predictor of scores on the authoritative subscale of the PSDQ at T1 ($R^2$ = 0.311, F(1, 16) = 7.57, $\beta$ = 0.605, p = .014), while emotional closeness was not retained as a significant predictor ($\beta$ = 0.166, p>.05). Analysis testing for prediction of scores on the authoritarian subscale of the PSDQ at T1 revealed that only the costs dimension was a significant predictor ($R^2$ = 0.385, F(1, 16) = 10.39, $\beta$ = -0.623, p = .005), while the child-SD interaction dimension was not ($\beta$ = -0.075, p>.05). Similar hierarchical linear regression analysis performed with the introduction of scores on all three dimensions of the MDORS at T2 at the second step for the prediction of scores on the Authoritarian subscale of the PSDQ at T2, revealed that, here too, only the costs dimension was a significant predictor ($R^2$ = 0.599, F(1, 16) = 24.08, $\beta$ = -0.797, p < .001), while other dimensions were not (respectively, $\beta$ = 0.028, $\beta$ = -0.214, both p>.05)

### 3.6. Association between changes in child's ASD symptoms and parent's well-being with changes in parenting strategies (Aim#4B)

No significant correlation was observed between changes in the ABI-S, PSI-S, STAI-Y-1 and Y-2 scores and changes in the authoritarian and authoritative subscale scores of the PSDQ, either when considering T1-T0 change scores (all r from 0.82 to 0.383 in absolute values, all p>.05) or T2-T0 changes (all r from 0.12 to 0.336 in absolute values, all p>.05). However, T1-T0 changes in the permissive subscale of the PSDQ correlated significantly with T1-T0 changes in ABI-S scores (r = .471, p = .035) and in STAI-Y-2 scores (r = .550, p = .012); while correlation with changes in PSI and STAI-Y-1 scores were not significant (respectively, r = .383, r = .121, both p>.05). Similarly, T2-T0 change scores on the permissive subscale of the PSDQ correlated positively with T2-T0 changes in PSI (r = .525, p = .017) and STAI-Y-2 scores (r = .452, p = .045); while no significant correlation with changes in ABI-S and STAI-Y-1 scores were observed (respectively, r = .383, r = .121, both p>.05). Thus, between T0 and T1, a decline in children's ASD symptoms and parents' trait anxiety was associated with a decrease in parents' use of permissive strategies. Conversely, between T0 and T2, a reduction in parents' parenting stress and trait anxiety was associated to a decrease in their use of permissive strategies.

Hierarchical linear regression analysis performed with the introduction of change scores T1-T0 on the ABIS-S and STAI-Y-2 at step two revealed that only scores on the STAI-Y-2 significantly predicted change scores T1-T0 on the permissive subscale of the PSDQ ($R^2$ = 0.138, F(1, 16) = 7.62, $\beta$ = 0.403, p = .014), while scores on the ABI-S did not ($\beta$ = 0.062, p>.05). Only change scores T2-T0 on the PSI-SF significantly predicted change scores T2-T0 on the permissive subscale of the PSQD ($R^2$ = 0.309, F(1, 16) = 7.34, $\beta$ = 0.567, p = .015), while scores on the STAI-Y-2 were not retained in the model ($\beta$ = 0.199, p>.05).

## 4. Discussion

Previous studies have demonstrated the numerous benefits that SD integration can have for children with ASD and their families. The aim of the present study was to propose a more systemic perspective on the effects of SD integration in families of children with ASD, through a longitudinal survey of 20 parents of children with ASD. Consistent with previous studies, we observed that parental well-being (i.e., parenting stress and anxiety) and children's ASD symptoms significantly improved after SD integration. Then, results showed that parents' well-being varied according to their child's symptomatology, and that changes in the child's ASD symptoms after SD were accompanied by changes in their parents' well-being. However, these benefits appeared to vary in relation to the quality of the relationship between the child with ASD and his/her SD. Finally, the results highlight that parenting strategies are associated with

the quality of their child's relationship with the service dog and that parents seem to adapt their parenting strategies according to the benefits of SD integration on both the child with ASD and the parent.

Current literature points out that parents of children with ASD experience higher levels of stress, reduced quality of life and are at higher risk for psychological health issues [24–28, 88], which have been shown to be particularly related to the severity and symptomatology of their child's ASD [31, 33]. The results of the present study confirmed this relationship between children's ASD symptoms and parental well-being (i.e., parenting stress and anxiety), as higher expression of ASD symptoms was associated with increased parenting stress and parental state anxiety. Parenting stress corresponds to the distress or discomfort that parents may experience in relation to their ability to meet childrearing demands or cope with parenting stressors [26, 92]. Parent-child interaction and the child's difficult/problematic behaviors are two elements that may be highly affected depending on ASD symptomatology, and these factors may serve as major stressors contributing to parenting stress [32, 33]. Additionally, ASD symptoms have numerous consequences on parents' daily lives (e.g., need for planification, respect of child's rigidity, need to anticipate potential tantrum generators and manage the child's tantrums) and require a great deal of attention from parents on a daily basis. Hence it is not surprising that ASD symptoms, which may affect parent-child interaction and contribute to the child's problem behaviors, are associated with parenting stress and anxiety.

Consistent with previous studies, the results confirmed that the integration of a SD benefits both the children with ASD and their parents [61–63, 67–72]. Significant improvements in core ASD symptoms were observed as early as 3 months after SD integration, while significant decreases in parents' trait anxiety and parenting stress were observed 6 months after SD integration. These findings not only confirm that the integration of a SD in the daily life of children with ASD has functional impacts for these children [61, 63, 64, 67, 69, 70], but also that these effects extend to their parents [29, 67–69, 71, 72]. A decrease in the parents' trait anxiety may appear surprising, as it is conceptualized as a stable individual difference in the propensity to feel anxiety. However, various studies have shown that a decrease in trait anxiety may be observed after different types of intervention [e.g., 99–101]. Furthermore, consistent with the higher prevalence of psychological health issues in parents of children with ASD [25, 26], we observed that of the twenty parents who participated in the present study, 19 had a total score above the clinical threshold on the PSI-SF (<90) and 19 had scores above the thresholds on both subscales of the STAI-Y (<40) simultaneously [92, 93]. Interestingly, some parents' anxiety and parenting stress scores fell below the clinical threshold after SD integration. Indeed, 6 months after SD integration the number of parents with scores above the clinical threshold decreased to 15 parents for the PSI-SF, and 14 for the STAI-Y (details in supporting information S2 Table). More generally, these results also seem to highlight that the integration of a SD can lead to significant benefits within a relatively short time since, as in previous studies based on a longitudinal methodology [29, 65, 70–72, 102], significant improvements can be observed within the first months after the arrival of the SD in the family.

Among the different hypotheses regarding the origin of these effects on parents of children with ASD, it has notably been proposed that the benefits of SD integration in children with ASD (e.g., decreases in tantrums, sleep problems and running away, increases in prosocial behaviors and social reciprocity), could lead to enhanced quality of life and reduced stress for parents [29, 71, 72]. Consistent with this hypothesis, we observed that the decrease in ASD symptoms following SD integration was associated with improvements in parental well-being (i.e., parenting stress and anxiety decreased). Thus, in relation to the previously mentioned association between ASD symptoms and parental stress/anxiety, it seems that the decrease in children's ASD symptoms after SD integration could lead to a decrease in the sources of

parental stress and anxiety. This last hypothesis seems particularly consistent with parental reports collected through interviews reported in previous studies [e.g., 68].

Several authors have hypothesized that the interaction between the child with ASD and the SD and the quality of the relationship they build may be at the core of the benefits of SDs for children with ASD [63, 73–76]. For instance, in terms of the benefits on the child's socio-emotional development, it has been argued that through the interaction with the SD, children with ASD could practice and refine their interaction skills and then generalize/transfer them to human interactions [64, 75, 103, 104]. The results of the present study seem to confirm the key role that the attachment between the child with ASD and the SD may play in the benefits of SD integration, as a stronger attachment between the child and the service dog, notably the child's emotional closeness with the SD, was related to a greater reduction in ASD symptoms. These findings are consistent with what has been previously reported in studies on pets in families of children with ASD using cross-sectional designs [74, 75, 77, 78]. Therefore, the ability to establish a qualitative relationship with the SD may be of critical importance for the child with ASD to achieve the desired benefits. In their process of attributing a service dog to families of children with ASD, organizations such as the Mira Foundation and the Handi'Chiens Asssociation are very attentive to the pairing of the child and the SD (e.g., compatible personalities, the SD shows behaviors adapted to the child's specificities, the child shows attraction to the SD) in order to ensure optimal chances that both will build a significant relationship. This attention to pairing seems relevant in order to ensure that the integration of their SD leads to the expected benefits, and seems even more important considering that not all children with ASD show the same interest toward SDs and not all develop a significant relationship with their SD [73–75, 78, 80, 104]. Various factors, including age and the severity of ASD, have been identified as contributing to the variability in the interest and interaction of children with ASD toward animals [80, 105]. For example, in their study on pets in the families of children with ASD, Grandgeorge et al. [80] found that children with ASD who experienced the arrival of a pet in their homes at 4–5 years of age developed a stronger relationship with their pet, and show a significant improvement in their prosocial behaviors; which was not the case for children with ASD whose family owned a pet prior to their birth.

The improvements in parental well-being, particularly the decrease in parenting stress, appear to be related to the quality of the relationship between their child with ASD and the SD, especially the emotional closeness. Two non-mutually exclusive hypotheses may be invoked to explain this result. First, a stronger child-SD relationship may result in stronger benefits on the development and symptomatology of children with ASD, which in turn may reduce the amount of potential stressors and alleviate some of the daily challenges for parents, all leading to an improvement in their quality of life. However, an alternative hypothesis which could be considered is the potential additional burden that the integration of a SD may involve in the case of a poor relationship between their child and the SD. Indeed, if the child with ASD does not become autonomous in the management of his/her SD and/or has little interest in handling and interacting with his/her SD, then the responsibility of caring for it falls onto the parent(s); which could be a source of additional burden for him/her/them [63, 76, 85]. Furthermore, we cannot exclude the possibility that children with ASD may have difficulties displaying appropriate behaviors toward the SD when the quality of the relationship is poor, which will require increased parental vigilance and monitoring of child-SD interactions. As mentioned in the participants subsection, one dyad had to return the SD to the non-profit organization because the child manifested inappropriate behaviors toward the SD. Consistent with testimonies mentioned in previous studies [e.g., 76, 84], during an interview conducted with that specific mother, she explained that she was concerned that her son would unintentionally hurt or cause discomfort to the SD. She even stated that because of her concerns for

the well-being of the SD, she would constantly watch and supervise any time her son was in the same room with the SD, which was a great source of fatigue and additional stress for her.

Regarding parenting strategies, the results at both follow-ups highlight that the quality of the child-SD relationship was associated with parenting strategies after SD integration. More specifically, the higher the quality of the relationship between the child and the SD, particularly in terms of the costs of the relationship, the less parents relied on authoritarian parenting strategies. Similarly, it seems that improvements following SD integration were related to changes in parenting strategies. In fact, we observed that the benefits of the SD, especially on the parent of the child with ASD (i.e., reduced parenting stress and trait anxiety), were notably associated with a decrease in reliance on permissive strategies. Interestingly, authoritarian parenting style has been pointed out as being more prevalent among parents of children with ASD [27, 54, 55], while both permissive and authoritarian parenting styles have been shown to have negative consequences for the child's development [42, 44]. Previous studies have reported on the association between parental stress and parenting strategies [45, 47, 48, 51]. For instance, in their study on NT children, Anthony and collaborators [46], found that the level of parenting stress was positively related to parents' reliance on discipline and negatively related to nurturance. Thus, the benefits of integrating a SD may contribute to an adjustment in their parenting strategies. Similarly, we cannot rule out that improvements in the child's ASD symptoms could also contribute to this adjustment in parenting strategies. Therefore, two non-exclusive hypotheses can be proposed regarding how improvements in ASD symptoms after SD integration may lead to changes in parenting strategies: either directly (i.e., parents adjust how they interact with their child according to how their child's symptomatology evolve) or indirectly (i.e., changes in the child's symptoms have an effect on parents' stress which in return has an effect on how they interact with their child). Finally, this study is the first to demonstrate that parenting strategies—particularly permissive and authoritarian strategies—can change following the integration of an SD and that these changes vary according to the benefits on the child and the parent. Due to the potential effects of parents' parenting strategies on children's development [42, 44], it could be assumed that these changes in parenting strategies (e.g., decreases in authoritarian and/or permissive strategies) could, in turn, be beneficial for the child with ASD and contribute to improvements in his/her development, in a kind of virtuous circle initiated by the dog.

## 5. Limits and future directions

The final sample size considered for analysis in the present study included a relatively small number of participants. A high attrition rate was observed, with fifteen parents not completing the survey at all three stages and thus not included in the analysis. High attrition rates are common in longitudinal studies [106]. Although we were able to show significant associations, this small sample size limited the ability to control for the effects of some confounding variables (e.g., child's age, presence of comorbidities, severity of ASD symptoms) as well as the ability to perform more complex analyses, such as multiple mediation analysis with inclusion of different mediators while controlling for the effect of different potential confounding variables, in order to further explore how all considered parameters may influence one another. More generally, replication of this study with a larger sample size would allow a stronger conclusion to be drawn, particularly regarding the hypothesis on the dynamics of effects proposed in the conclusion section. Additionally, increasing the sample size would also allow the generalizability of this finding to be explored, by including a greater diversity of children within the spectrum, for example. Moreover, replication with the inclusion of a control group consisting, for example, of families of children with ASD on the waiting list for a SD, with follow-ups over a

similar period of time, should strongly be considered for future studies in order to perform comparisons and avoid potential bias effects.

One of the strengths of the present study is that we did not consider the effect of SDs provided by a single source, but of SDs provided by two different organizations. Future studies focusing on the effects of SDs on children with ASD and their families would benefit from including a large panel of organizations providing SDs using different procedures and breeds.

Recently, Guay and collaborators [85] showed that families of children with ASD were significantly more satisfied with having a SD than families with a companion dog. Thus, future studies could benefit from replicating the present study with families that have integrated a companion dog in their household. The inclusion of such a sample would enable a more in-depth exploration the specific effects of SDs. Similarly, it might be of interest to consider other pet species in order to observe whether some benefits are specific to dogs.

Due to its multi-dimensional nature, well-being is a complex concept that encompasses numerous elements, such as feelings of competence, self-esteem, positive emotions and resilience. However, in the present study, the effects of SDs on parental well-being were considered solely through their impact on parental anxiety and parenting stress. Therefore, future studies could consider expanding the scope of investigation by exploring the effects of SDs on other aspects of well-being and/or by using multidimensional measures of well-being, such as those developed by Huppert and So [107]. Additionally, performing a psychiatric evaluation on parents prior to the integration of the SD, could have been of interest. Indeed, knowing parents of children with ASD are at higher risks of psychological health issues, notably anxiety and depression, compared to parents of NT children [24–27], it would have been of interest to perform such type of evaluation in order to explore if parents initial mental health status was related to their child with ASD's symptoms and to their parenting strategies, as well as with the different benefits and changes observed after SD integration.

In a recent study, Carlisle and collaborators [74] demonstrated that parents' perception of the benefits of pet ownership was positively related to the bond between their child with ASD and their pet, but also to their own bond with it. Hence, future studies could explore the parent-SD relationship in order to account for the full range of potential reciprocal influences within the child-parent-SD trio. With the same objective in mind, it would also be valuable to gather data on the well-being of the SD. Indeed, it is to be expected that the quality of the child-dog relationship, or even the severity of the ASD (particularly the expression of problematic behaviors), may have consequences on the dog's well-being. For example, based on interviews of parents of children with ASD, Burrow and collaborators [82] identified child's meltdowns, aggressive behaviors, intrusive touching, solicitations and the predictability of the child's behavior as potential stressors that may affect the well-being of the SD.

The present study highlights that integrating a SD within the family of a child with ASD may lead to dynamic and reciprocal influences between the child with ASD and his/her parent. Completion of the survey at each stage was performed by only one of the child with ASD's parents, usually the mother. This methodological choice was mainly motivated by a concern for balance between single-parent and two-parent families. However, within a same family, the two parents may present different parenting styles. Future research should thus consider collecting data on both parents, when possible, in order to consider the ecosystem of parenting styles within the environment of the child with ASD. Additionally, and more generally, extending the investigation of the benefits and influences of SD integration to other family members (e.g., second parent, siblings) could be relevant for future studies in order to explore the effects of a SD on the entire family microsystem it becomes a part of. Taking into account the whole family system seems even more pertinent when one considers that having a pet in NT families can increase family adaptability and cohesion [108]. In a recent qualitative study

of families of children with ASD with SDs, Leighton and collaborators [109] demonstrated that the presence of a SD contributes to strengthening and stabilizing the family unit, and improves the social functioning of the family unit (i.e., the interaction of the family with external systems such as other families, communities, schools, etc.). Thus, in order to further our understanding of the effects of SD integration on the family system, it would be of great interest to conduct studies that examine its effects from the family micro-system (i.e., family unit) to the family meso-system (i.e., peers, friends, school, community, etc.). By exploring the effects of SD integration on the social perception, judgement and social participation of children with ASD and their family, for instance, a more comprehensive understanding can be gained.

## 6. Conclusion

To the best of our knowledge, this study is the first to quantitatively investigate the reciprocal influences between the benefits of SD integration on children with ASD and their parents, as well as the first to explore the association of these benefits with parenting strategies and the child-SD relationship, using a longitudinal approach. Based on the results and associations observed in the present study, we may formulate a hypothesis regarding the dynamics of the effects that SD integration in families of children with ASD may trigger. First, it seems that integrating a SD will have benefits on the child and his/her ASD symptoms. However, those benefits seem to rely on the interaction and attachment between the child and the SD, and thus vary according to the quality of the child-SD relationship. Then, the benefits on the child with ASD and the decrease of his/her symptoms will in turn lead to benefits on the parents, and especially lead to a decrease in their parenting stress and anxiety. In return, those benefits for parents and on their quality of life lead to changes and adaptations in their parenting strategies, which may themselves be beneficial to the development of their child with ASD.

More generally, by considering the effects of SDs for youth with ASD from a more systemic perspective, this study not only confirms that the integration of a SD has beneficial effects on both the child with ASD and his/her parent, as shown in previous studies, but also highlights the dynamic and reciprocal nature of these benefits for the child and the parents (i.e., benefits on the child are associated with benefits on parents which in turn may improve the parents' interaction strategies with their child). It also shows for the first time that parents of children with ASD may adapt/change their parenting strategies in response to the improvements that integration of the SD may induce, thus demonstrating a certain flexibility in parents' parenting behaviors and strategies. Finally, it emphasizes the central role that the relationship between the child with ASD and the SD may play in the observed benefits.

## Supporting information

**S1 Table. Additional information relative to the standardized scales and questionnaires used in the present study.**
(DOCX)

**S2 Table. Raw data on the final sample on all standardized scales and questionnaire at each follow-up.**
(DOCX)

## Acknowledgments

We express our heartfelt gratitude to the Mira Foundation (especially, Nicolas Saint-Pierre, Eric Saint-Pierre, Noël Champagne, Catherine Deschatelets, Fanny Kearnan, Mylène

Chaumette, Aurélie Tremblay) and the Handi'Chiens Association (especially, Alain Legrand, Marie-Claude Lebret, Nathalie Favier-Hannequin, Eve Janodet, Sophie Mary, Emilie Ladrat, Sophie Collin, Marine Marie, Charlyne Eury) for their invaluable help and collaboration in conducting this study. We are also grateful to all the members of the guidance committee of the MECA_TSA project for their valuable assistance and guidance throughout every step of this research project. We extend our special thanks to the staff and professionals from the Handi'Chiens Association and the Mira Foundation for their dedicated efforts and assistance in recruiting participants. Finally, we would like to extend our most sincere acknowledgements to all the families and children with ASD who graciously agreed to participate in the present study. Their willingness to contribute has been invaluable to our research.

## Author Contributions

**Conceptualization:** Nicolas Dollion, Florian Auffret, Nathe François, Pierrich Plusquellec, Marine Grandgeorge.

**Data curation:** Nicolas Dollion, Margot Poirier, Marine Grandgeorge.

**Formal analysis:** Nicolas Dollion, Margot Poirier, Marine Grandgeorge.

**Funding acquisition:** Nicolas Dollion, Marine Grandgeorge.

**Investigation:** Nicolas Dollion, Margot Poirier, Marine Grandgeorge.

**Methodology:** Nicolas Dollion, Margot Poirier, Florian Auffret, Nathe François, Pierrich Plusquellec, Marine Grandgeorge.

**Project administration:** Nicolas Dollion, Florian Auffret, Nathe François, Pierrich Plusquellec, Marine Grandgeorge.

**Resources:** Nicolas Dollion, Florian Auffret, Nathe François, Marine Grandgeorge.

**Supervision:** Nicolas Dollion, Florian Auffret, Nathe François, Pierrich Plusquellec, Marine Grandgeorge.

**Writing – original draft:** Nicolas Dollion.

**Writing – review & editing:** Nicolas Dollion, Margot Poirier, Florian Auffret, Nathe François, Pierrich Plusquellec, Marine Grandgeorge.

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
