## [Decision Letter · Decision Letter 0]

19 Oct 2023

PONE-D-23-30617Effects of service dogs on children with ASD’s symptoms and parents' wellbeing: on the importance of considering those effects with a more systemic perspective.PLOS ONE

Dear Dr. Dollion,

Thank you for submitting your manuscript to PLOS ONE. After careful consideration, we feel that it has merit but does not fully meet PLOS ONE’s publication criteria as it currently stands. Therefore, we invite you to submit a revised version of the manuscript that addresses the points raised during the review process.

The two reviewers addressed several major concerns about your manuscript. Please revise your manuscript according to reviewer's comments.

We look forward to receiving your revised manuscript.

Kind regards,

Kenji Hashimoto, PhD

Section Editor

PLOS ONE

Journal Requirements:

3. Please upload a copy of Supporting Information Figure/Table/etc. Supplementary Material: Table A and Table B which you refer to in your text on pages 46 and 47.

Reviewers' comments:

Reviewer's Responses to Questions

**Comments to the Author**

1. Is the manuscript technically sound, and do the data support the conclusions?

Reviewer #1: Partly

Reviewer #2: Partly

2. Has the statistical analysis been performed appropriately and rigorously? 

Reviewer #1: I Don't Know

Reviewer #2: N/A

3. Have the authors made all data underlying the findings in their manuscript fully available?

Reviewer #1: Yes

Reviewer #2: Yes

4. Is the manuscript presented in an intelligible fashion and written in standard English?

Reviewer #1: No

Reviewer #2: Yes

5. Review Comments to the Author

Reviewer #1: No specific comments to the authors. See my letter below to the editor.

xxxx xxxx xxxx xxxxx xxxx xxxx

wwwwwwwwwwwwwwwwwwwww

eeeeeeeeeeeeeeeeeeeeeeeeeee

rrrrrrrrrrrrrrrrrrrrrrrrrrrrrrrrrrrr

111111111111111111111111111

222222222222222222222222222

333333333333333333333333333

4444444444444444444444444444

555555555555555555555555555555

66666666666666666666666666666666

88888888888888888888888888888

99999999999999999999999999999

111111111111111111111111111111111111111111111111111111111111111111111

222222222222222222222222222222222222222222222222222222222222222222222

333333333333333333333333333333333333333333333333333333333333333333333333333

444444444444444444444444444444444444444444444444444444444444444444444444

555555555555555555555555555555555555555555555555555555555555555555555555

666666666666666666666666666666666666666666666666666666666666666666666666

777777777777777777777777777777777777777777777777777777777777777777777777

888888888888888888888888888888888888888888888888888888888888888888888

Reviewer #2: The authors stated that 'To the best of our knowledge, this study is the first to quantitatively investigate the reciprocal influences between the benefits of SD (Service Dog) integration on children with ASD (Autism spectrum disorder) and their parents, as well as the first to explore the association of these benefits with parenting strategies and the child-SD relationship, using a longitudinal approach. '

This study is very interesting. However I have a few questions.

#1: Would you tell me the definition of parenting strategies in this study?

#2: Related to #1, has a psychiatric evaluation been done on the parents?

6. PLOS authors have the option to publish the peer review history of their article (what does this mean?). If published, this will include your full peer review and any attached files.

Reviewer #1: No

Reviewer #2: No

---

## [Author Response · Author response to Decision Letter 0]

15 Nov 2023

Detailed answers to Academic editor's Notes and Revierwers' comments are available in the document entitled "Rebuttal letter".

Response to academic Editor Notes:

In addition to the rebuttal letter, two versions of the manuscript have been generated and submitted on the submission platform: one corresponding to the revised version with all changes made highlighted in yellow ('Revised Manuscript with Track Changes'); the second corresponding to the unmarked revised version (‘Manuscript’).

No changes to our initial financial disclosure had to be performed.

We ensured that the revised manuscript meets all Plos One’s style requirements, and that file naming respects Plos One’s guidelines.

Our study’s minimal underlying data set was previously directly embedded with the manuscript as Supplementary material Table B. In this revised version of the manuscript, the minimal data has been integrated as Supporting Information in a separate file (S2 Table).

In the revised version of the manuscript, Table A and B are integrated as Supporting Information (S1 Table and S2 Table) in separate dedicated files.

Response to reviewers' comment:

Reviewer#2 referred "No" in answer to "4. Is the manuscript presented in an intelligible fashion and written in standard English?"

English has been fully revised by an English native professional.

Reviewer#1 referred "Would you tell me the definition of parenting strategies in this study?"

We thank reviewer #2 for this comment. “Parenting strategies” was not clearly defined in the initial version of the manuscript. Additionally, this comment allowed us to notice that in the initial version of the manuscript sometimes the term “parenting profile” was used and other times “parenting style” was used (which in fact are common expressions referring to the same concept in the scientific literature), which could have increased the confusion to the reader on the difference between parenting style and parenting strategies. 

The definition of parenting strategies is now clearly stated, and only the expression “parenting style” used. In line with those modifications, we also adjusted the presentation of the PSDQ (P4-5 and P.14)

Reviewer#1 referred "Related to #1, has a psychiatric evaluation been done on the parents?"

We did not perform a psychiatric evaluation on parents. However, it would have indeed been of interest in order to draw additional conclusion; notably to explore if parent’s depression and anxiety disorder is related to variables prior to SD integration (i.e., child’s ASD symptoms, parenting strategies), and to changes after SD integration. We added this element in the “5. Limits and future directions” section (third paragraph)(P.33-34)

---

## [Decision Letter · Decision Letter 1]

28 Nov 2023

Effects of service dogs on children with ASD’s symptoms and parents' well-being: on the importance of considering those effects with a more systemic perspective.

PONE-D-23-30617R1

Dear Dr. Dollion,

We’re pleased to inform you that your manuscript has been judged scientifically suitable for publication and will be formally accepted for publication once it meets all outstanding technical requirements.

Kind regards,

Kenji Hashimoto, PhD

Section Editor

PLOS ONE

Additional Editor Comments (optional):

Reviewers' comments:

Reviewer's Responses to Questions

**Comments to the Author**

1. If the authors have adequately addressed your comments raised in a previous round of review and you feel that this manuscript is now acceptable for publication, you may indicate that here to bypass the “Comments to the Author” section, enter your conflict of interest statement in the “Confidential to Editor” section, and submit your "Accept" recommendation.

Reviewer #1: All comments have been addressed

Reviewer #2: All comments have been addressed

2. Is the manuscript technically sound, and do the data support the conclusions?

Reviewer #1: Yes

Reviewer #2: Yes

3. Has the statistical analysis been performed appropriately and rigorously? 

Reviewer #1: Yes

Reviewer #2: N/A

4. Have the authors made all data underlying the findings in their manuscript fully available?

Reviewer #1: Yes

Reviewer #2: Yes

5. Is the manuscript presented in an intelligible fashion and written in standard English?

Reviewer #1: Yes

Reviewer #2: Yes

6. Review Comments to the Author

Reviewer #1: The paper is interesting and the methodology sound - especially as the authors include a brief discussion of the limitations of their study.

Reviewer #2: Thank you for your polite response about below.

#1: Would you tell me the definition of parenting strategies in this study?

#2: Related to #1, has a psychiatric evaluation been done on the parents?

7. PLOS authors have the option to publish the peer review history of their article (what does this mean?). If published, this will include your full peer review and any attached files.

Reviewer #1: No

Reviewer #2: No

---

## [Editor Report · Acceptance letter]

6 Dec 2023

PONE-D-23-30617R1 

Effects of service dogs on children with ASD’s symptoms and parents' well-being: on the importance of considering those effects with a more systemic perspective. 

Dear Dr. Dollion:

I'm pleased to inform you that your manuscript has been deemed suitable for publication in PLOS ONE. Congratulations! Your manuscript is now with our production department. 

Kind regards, 

on behalf of

Prof. Kenji Hashimoto 

Section Editor

PLOS ONE